

# DESIGN OF 24X7 WATER SUPPLY SYSTEMS

*A Case study: Ahmedabad city*

[1]Swarup Varu and [2]Dipsha Shah

[1]Graduate Student, [2]Professor
School of Building Science & Technology,
CEPT University, Ahmedabad, India

_________________________________________________________________________________________

*Abstract:* Water found one of the important physical environments of human and has a direct behavior on the health and hygiene of mankind. There is no denying the fact that the contamination of water leads to numerous health hazards. The facility of safe and adequate drinking water to the growing urban population continues to be one of the major challenging tasks for any state. In India, according to the Ministry of Urban Development (MoUD); continuous potable water supply to every households are directly related with the Service Level Benchmark (SLB) of the city.

This study is applied research and it designs and builds a detail project report for 24x7 water supply system at Sabarmati and old Wadaj ward of Ahmedabad city. It includes feasibility study; software based hydraulic design, operation and maintenance strategy and economic feasibility for the project by studies of research paper, case study, census data, need and demand of the future. The present water supply practice is non-confirming to designed hydraulic parameters, and also the system is severely affected insufficient hydraulics, leading to many of the current critical issues which keep the Local Authorities in an evitable brutal circle. Using data from the local government body, the papers presents the condition of intermittent water supply network and compare it with continuous water supply system of selected District Metering Area (DMA) of Ahmedabad in Gujarat (India).

*Index Terms* - **Continuous water, 24x7 water supply system, Economic Feasibility, telescopic tariff, smart city.**

_________________________________________________________________________________________

## I. INTRODUCTION

The decreasing availability of water supplies is one of the most important environmental issues taken by various countries. In many urban areas, intermittent service, wherein water is provided to residents for a limited number of hours per day. The term continuous water supply – refer to the supply of potable water to end users through a system of pipes-covering interlinked bulk transmission and/or distribution system which are continuously full and under positive pressure throughout their whole length, such that the end user may draw off water at any time of the day or night, 24x7 throughout the year. This is by itself an important aim for any water supply system. Continuous supply has two main advantages. One is that people can draw water when they need. The second advantage is not holding contamination. When pipes are empty most of time, contamination can seep in through cracks and gaps. A pipe carrying 24 hours a day, on the other hand, will not allow this as the water pressure is acting centrifugally.

It is internationally acknowledged that the best way to keeping water safer during distribution is to ensure that it keeps flowing through the pipes on a regular (24x7) basis. Due to irregular or intermittent water may get contaminated due to pressure difference in pipe or stagnation of water which ultimately leads to health hazard. An intermittent water supply is common to most of the Indian cities. The consumers are forced to collect as much water as possible during the limited supply hours, which leads to excessive usage of pumps.

Continuous water supply is one of the important norms for smart cities. Recently the Government of India has decided to convert 100 cities into smart cities. Intermittent water supply is one of the impediments that get in the way of not only improving service delivery but also for conversion of a city to a smart city. There have been many attempts to achieve this most challenging task. Unplanned distribution system and laxity in water loss management makes 24x7 water supply a difficult task. "The Manual of Water Supply & Treatment” 3rd Edition CPHEEO, Ministry of Urban Development, New Delhi: stats that,

*"The intermittent system suffers disadvantages, wherever possible, intermittent supply should be discouraged or convert it into the continuous supply system for better health and wealth of the citizens"*

The objective of this paper is to make a successful project design for the 24x7 water supply system in Ahmedabad ward.

## II. RESEARCH DESIGN

Detail research methodology for the 24x7 water supply project in Ahmedabad ward level is as described in Figure: 1



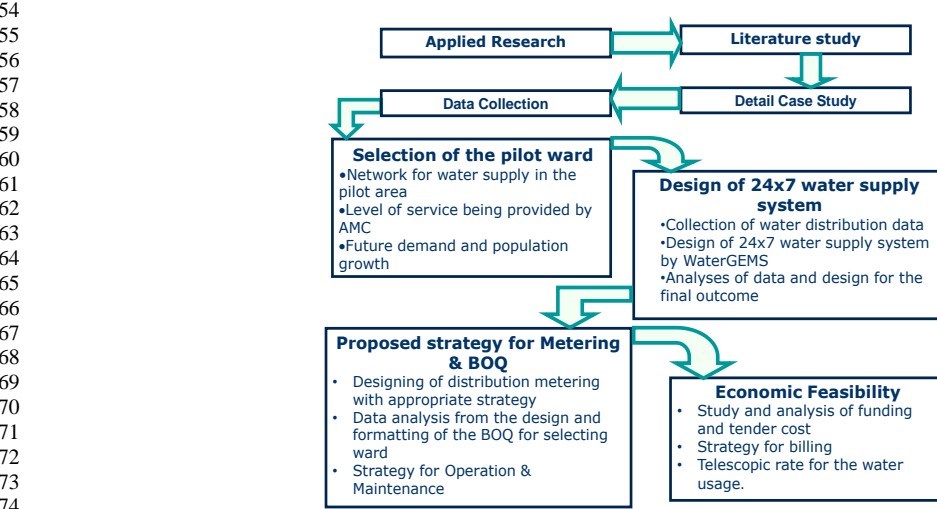

*Figure 1: Research Methodology*

## III. CASE STUDY

The scheme for providing continuous water supply in Malkapur, Maharashtra in India is unique and noteworthy as it has been implemented on a town-wide basis. A range of systematic interventions – management processes human resource development, new technologies, policy and financial measures, citizen friendly services, services to the poor, better coverage, communication – were effectively employed and the process changes were institutionalized. More importantly, these innovative practices have demonstrated that providing continuous water supply is an achievable objective. The initiative has substantially improved service delivery standards for the public health benefits.

Integrating geographical information system (GIS) based maps with household survey data and hydraulic model in WaterGEMS, HDPE pipes material for distribution pipe, automated meter reading system has been another innovative intervention which has made spatial information available for better decision making.

It has been observed in continues water supply system that more than 60% households have a bill below Rs.100/month. The rate is higher for commercial connection such as hotel/business etc. The financial statement of the revenue and expenses for Malkapur has shown that the scheme is financially Viable.

Maharashtra Jeevan Pradhikaran (MJP) has conceptualized, designed and implemented the initiative in-house. Internal capacities have been nurtured carefully at all levels and team work was promoted. Extensive engagement with all levels of stakeholders, i.e. political, officials, citizens, etc., ensuring their support for the initiative. Effective and relevant use of technology was promoted by MJP. The initiated process improvements were institutionalized. The initiative has demonstrated that a small town with limited resources can successfully implement and run a continuous water supply system. The commitment of political leadership was noteworthy. The initiative was implemented in-house and town-wide while other towns have relied on private expertise to implement continuous water supply on a pilot basis.

## IV. STUDY AND FINDING

It's the critical responsibility of the Local Body to supply 125-150 liters of water per head per day per household 24x7, that too with adequate pressure. However, to cope up with the demographic growth of the urban centers, Local bodies adopted the strategy of limiting the supply period. Now almost all local authorities resort to cut supply to an extent of few hours or even few minutes for their city supplies. Also, Water supply of Ahmedabad (Gujarat) is not as per the required norms. Though there is adequate raw water and treatment plant facility is available, there is lack of effective water supply system result in poor satisfaction level among the users. At present, water supply department supplies the water on basis of time of supply other than demand of users as per standards of CPHEEO manual (Gov. of India). As a result of such practice of shortened water supply from the authorities, people try to preserve as much water that is made available to them during the supply time. This obviously results into excessive inflection of water by every household, with their taps fully open during the supply period.

This practice harmfully affects the hydraulics of the pipe network. In fact, people draw water simultaneously, store in their household, and consume for 24 hours the water quantity that has been made available in short supply period. For this, they need to store, and also expend electrical energy for lifting the water to the elevated tanks of every individual household. In short, during the shortened supply period, there is simply transfer of water storage from municipal storage to the individual household.



As per the guideline note for continuous water supply by Ministry of urban development, Government of India, Area has
been divided into small District Metering Area (DMA). District Metering Areas are the basic building blocks (Wards) of zones
distribution system which provide a manageable unit by which distribution consumers and performance information can be
achieved. From the 6 zones of Ahmedabad city West Zone has highest percentage of population and area covered by water
distribution network and among all wards of the west zone Sabarmati and old Wadaj has maximum coverage of population with
treated tap water within premises. Therefore, for the detail study of 24x7 water project, these two wards have been selected.
WaterGEMS software has been used for design hydraulic model of 24x7 water supply system. "WaterGEMS is a
hydraulic modeling application for water distribution system with advanced interoperability, geospatial model building,
optimization, and asset management tools. WaterGEMS includes ArcGIS and AutoCAD."
Design Process for 24x7 water supplies in WaterGEMS software in Figure.2

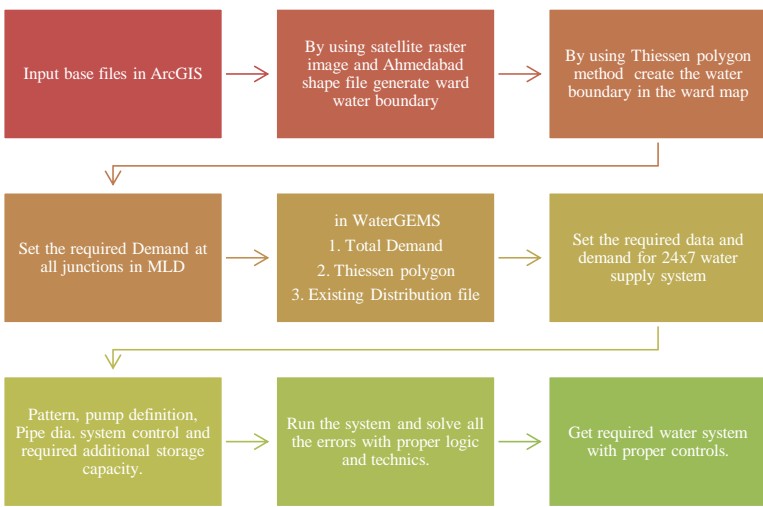


*Figure 2: Design Process in WaterGEMS*


## V. DATA COLLECTION AND DESIGN



Data collection has been done from census data and from local Government water supply department for selected DMA.
From the design, it was founded that to convert intermittent to 24x7 water supply system, 85% of the existing network has to be
replaced. There are technically three possibilities for implementation of 24x7 water supply system, which are different in material
and construction technology shows in Figure: 3 & 4.

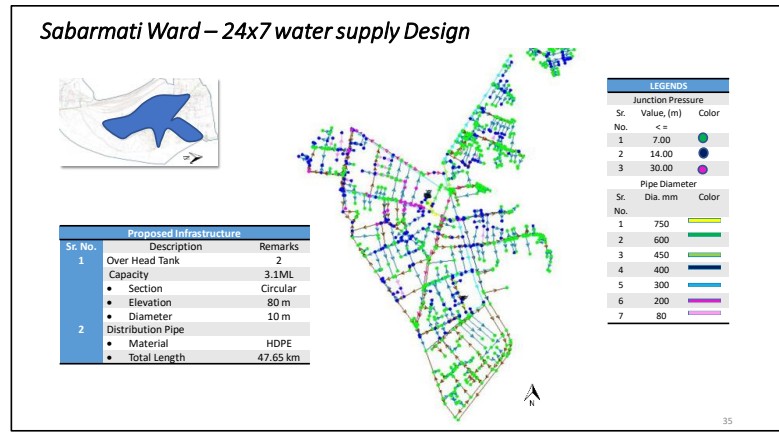


*Figure 3: 24x 7 water supplies design for Sabarmati ward*



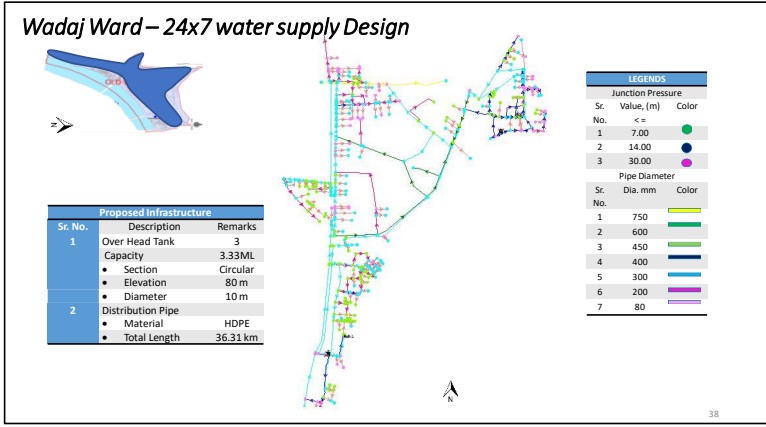



*Figure 4: 24 x7 water supply design for Old Wadaj ward*


## VI. COST ANALYSIS SCENARIO



For cost analysis, three scenarios have been selected according to possibilities. Each has its own advantages with respect
to cost, technology, environment, socially and economically.

1    85% Replacement of Distribution system with existing material and by using traditional excavation method.
(Total Project cost is fifty four cr.)

2    85% Replacement of Distribution system with existing material and by using Trenchless Technology.
3    (Total Project cost is fifty seven cr.)

4    New Distribution system with High Density Polyethylene HDPE pipes and Trenchless Technology. (Total
Project cost is seventy two cr.)

By considering social, environmental & sustainability of the project, option 3 has been selected for the 24x7 water
supply system in both the wards. Hence, Final project cost for the 24x7 water supply system in Sabarmati & Old Wadaj ward is in
Table: 1

Table 1: Total cost of the Project

| Sr. No. | Description | Total Cost, Cr. Rs |
|---|---|---|
| 1 | Total Cost of the Project in Sabarmati Ward | 38.87 |
| 2 | Total Cost of the Project in Old Wadaj Ward | 32.77 |
| **TOTAL PROJECT COST Rs.** ≈ | | **72.00** |


## VII. ECONOMIC FEASIBILITY



In India, revenues from water and sewerage services typically cover less than 30 percent of operating costs. As a result,
water and sewerage services have to be heavily subsidized by government. Main focus should be on better infrastructure that has
a direct link to provision of better services to people therefore, government of India has introduced AMRUT and Smart city
scheme. Ahmedabad has get Rs. 115 crore under AMRUT Scheme for water augmentation and water projects in financial year
2016-17.
For the better funding and to run project successfully it is recommended to implement project on PPP mode with 90%
Gov. Shares and 10% public shares. From the calculation and analysis, it has been founded that one time water meter installation
cost per house is Rs. 2,300 and monthly operation and maintenance charge for every house is Rs. 200. For usage of water, the
telescopic tariff has been design in 4 slabs according to usage of water in different premises like residential and commercial.
Telescopic tariffs are Rs.7.5 /kilo liter when the usage of the water is in between 13.5 to 20.25kilo liter (KL) per month; Rs.10

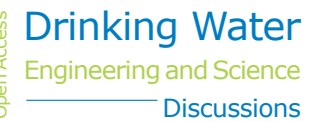

/KL when usage is in between 20.25 to 27KL per month; Rs.15 / KL when usage is in between 27 to 33.75 KL per month and
Rs.30 / KL when the usage of water is more than 33.75KL.
Revenue has generated from the connection, operation & maintenance charge and water tariff are as shown in the below
table, which is used to calculate the expense recovery ration of the project in Table: 2

*Table 2: Total Revenue generate from the project*

| Sr. No. | Description | Unite No. | Rate Rs. | Total Cost in Cr. Rs. |
|---------|-------------|-----------|----------|-----------------------|
| 1 | One time connection charge | 30,531 households | 2,300.00 | 70.22 |
| 2 | O & M cost | 30,531 households | 2,400.00 | 73.27 |
| 3 | Average water usage charge | 30,531 households | 21,972.00 | 16.77 |
| **TOTAL REVENUE COST  Rs.** | | | | **31.12** |

E/R ratio = $\frac{31.12}{64.80}$ x 100 = 48.05% which is < 100

(10% of total project cost is 7.2cr funding would be from public remaing 64.80 cr from GoI)

Therefore, E/R ratio of the project shows that the project is economically viable to execute and the project recovery is short term.

## VIII. CONCLUSION

Findings and conclusion from the research and analysis is as below:
1.   Ahmedabad city has highest need to convert its water distribution system from intermittent to 24x7 water supplies to
achieve **service level benchmark**.

2.   **West Zone** has highest coverage of area and population by water distribution network among all the 6 zones of
Ahmedabad; **Sabarmati and old Wadaj** are the two wards among the 10 wards of the west zone having highest
percentage of population cover with water distribution network and maximum households are get treated tap water
within their premises.

3.   Design of water distribution network, with help of the WaterGEMS for Sabarmati and old Wadaj ward shows that to
convert intermittent to 24x7 water supply system, around **85% of the existing network has to be replaced.**

4.   For the 24x7 water supply system, there is need to construct more elevated storage reservoir in both the ward,
**Sabarmati Ward: Total 3.1 ML capacity**
**Old Wadaj: Total 3.33ML capacity**

5.   There are technically three possibilities for implementation of 24x7 water supply system, which are different in material
and construction technology which shows in Table: 3

*Table 3: Scenario wise project cost*

| Sr. No. | Ward | Scenario 1 | Scenario 2 | % Increase | **Scenario 3** | **% Increase** |
|---------|------|------------|------------|------------|----------------|----------------|
| 1 | Sabarmati | 29.12 | 30.92 | 6.18 | **38.87** | **33.44** |
| 2 | Old Wadaj | 24.94 | 26.30 | 5.45 | **32.76** | **31.34** |

6.   By considering social, environmental & sustainability of the project, **scenario 3** has been selected for the 24x7 water
supply system in both the wards.

7.   Fund is distributed between Government and Public (**90:10**)



8. **One time water meter installation** cost per house is Rs. **2,300** and monthly operation and maintenance charge for every house is Rs. 200.

9. For usage of water, the telescopic tariff has been design in 4 slabs according to usage of water as described in Table:4

*Table 4: Telescopic tariff*

| Sr. No. | LPCD | Kilo Liter | Charge Rs /Kilo Liter | Range Rs / Month |
|---|---|---|---|---|
| 1 | 100-150 | 13.50-20.25 | 7.50 | **102-152** |
| 2 | 150-200 | 20.25-27.00 | 10.00 | **202-270** |
| 3 | 200-250 | 27.00-33.75 | 15.00 | **405-507** |
| 4 | >250 | 33.75 & More | 30.00 | **>1012** |

10. E/R ratio of the project is 48.05% which shows that the project is **economically viable** to execute and project recovery is **short term**.

24x7 water supply system is equally depended on technical, social, political and financial aspect of the project. This project is highly recommended better planning and operation strategy to run the project successfully. In that term, the research project in both the ward is viable by design aspect, techniques & economy.

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
