# Peer review of "DESIGN OF 24X7 WATER SUPPLY SYSTEMS"

_Drinking Water Engineering and Science, 2018_

## Referee Comment (RC1) · Anonymous Referee #1 · 18 Aug 2018

**Review Report**

**Paper Title: Design of 24x7 Water Supply Systems – A Case Study: Ahmadabad City**

**Authors: Swarup Varu, Dipsha Shah**

Conversion of existing intermittent mode of operation to continuous (24x7) mode requires ascertaining: (1) adequacy of water at the source; (2) adequacy of the network components to deliver desired quantity of water at required pressure; (3) effective operation and maintenance to minimize operational cost and leakages; and (4) proper water tariff, billing and collection system for self-sustenance of the water supply system. Even though consumer demands may be met in intermittent mode of water supply, the main disadvantages of intermittent mode as mentioned by authors also are: (1) Possibility of contaminant intrusion when pipeline is not under pressure; (2) Coping costs at households towards storage facilities, pumping of water, and household treatment facility; and (3) Possibility of contamination at individual storage tanks. The authors emphasized on design of 24x7 WSS in the case study presented by them. I have several points for the authors to address in the manuscript.

1. Residual pressure at different nodes is observed to vary between 7 to 30 m in both Sabarmati and Old Wadaj Ward. The residual pressure of 7 m may not be sufficient for supply of water directly to consumers located on second and higher floors. Why the network is not designed for higher residual pressures to reduce coping cost?

2. Section III – Case Study: It is only after reading the 4th paragraph of this section, I found that Case Study of Malkapur is conceptualized, designed by implemented by Maharashtra Jeevan Pradhikaran and authors were not involved with that study. Reference from where data and results have been reported should be provided at appropriate location in the text.

   What is the necessity of last line in the fourth para? What are those towns referred by "other towns"? What are the problems faced by other towns while implementing the scheme through private expertise? The present study by the authors also involves implementation under PPP mode!

3. Section IV – Study and Findings: Study findings should be supported with data using performance indicators like hours of supply; per capita supply; residual pressure, leakage and other losses; frequency of supply; mode of charging, expenditure and recovery, etc. of existing system. The last para about information on software used should be shifted in next section.

4. Section V – Data Collection and Design: This section should include important design criteria. Also, include whether Darwin Designer is used to get optimized sizes, or the same have been obtained by trial and error method. How the existing pipes have been considered in design? How the design can inform that 85% of existing pipes is to be replaced? What is the criterion used for pipe replacement? On what basis option of strengthening the network by parallel pipes is ignored? Figures 3 and 4 do not depict anything about three technical possibilities of implementation.

5. Section VI – Cost Analysis Scenario: Would it not be better to consider strengthening of existing network as one of the alternative? How option 3, which is the costliest, is advantageous from social, environmental and sustainable point of view? HDPE pipes as well as trenchless technology have their own disadvantages. Do the authors recommend such an alternative for all cities willing to change to 24x7 mode of water supply?

6. What is meant by public share? If I am not wrong, it is to be contributed by local body. If it is so, where is PPP?

7. There seems to be something wrong in presentation in Table 2. Is O&M cost proposed to be collected annually or onetime cost? What is the period of analysis considered for obtaining average water charges? How increase in household over period of design is accounted? How the cost is capitalized? Why the total sum is not correct? What is the meaning of recovery? Is it that full cost is recovered? No analysis for payback period is provides. How it can be considered short period.

8. Any project can be shown economically viable by deciding high tariff. Was there any study carried out for "Willingness to pay" for better services. Many apartments type of building have several households with single connection. How the tariff is decided for them?

9. Suggesting replacement of 85% of existing pipes by new pipes irrespective of age and condition of pipes as solution for design of 24x7 system cannot be accepted by any local body.

The paper is poorly written and provides nothing new as far as design is concerned. Even I don't feel anything worth documenting and useful for readers. Further, authors' suggestions have severe drawback. Pressure management, leakage control, reliability of the system are some of the concerns that need to be considered while designing 24x7 system.

---

## Referee Comment (RC2) · Anonymous Referee #2 · 21 Nov 2018

The paper aims at discussing the design of a water supply system in urban India that runs continuously, avoiding constraints in supply and water quality. However, the paper is poorly written, it lacks a literature overview of similar studies, a thorough discussion about the considerations during the design, and the advantages/disadvantages of the used approach. The results are not discussed in the light of similar studies. Specific comments: - Line 9, avoid very general statements - Line 19, avoid subjective statement (as "brutal circle") - Line 35, as in the rest of the paper give reference to this statement - Line 40, continuous water supply is not only important for "smart cities". - Line 52, describe in words what we see in Figure 1. - Line 77-98, not clear what is the purpose of the description of this case study. - Line 115, "Area has been subdivided. . ." which Area? - Line 245-247, general statement which is not the result of the study.

---

## Author Comment (AC1) · 14 Dec 2018

**Responses to the Editor's and Reviewers' Comments**

We appreciate very much the editor and the reviewers for the constructive comments. We also thank the editor and the reviewers for the effort and time put into the review of the manuscript. Each comment has been carefully considered point by point and responded. Responses to the reviewers and changes in the revised manuscript are as follows.

There is nothing about to emphasize 24x7 water supply system (WSS), it has its own advantages which is appreciated worldwide. And as I said in the paper this research is purely applied research so whatever advantages I have mention in the case study (Malkapur, India) all are proven after its detail study by government and concern authority, which is in cited references. So, not even a single point has been added to just emphasize the system.

**1.** We appreciate your detail comments. You are right that 7 m may not be sufficient for higher floor but here in design for both the ward Sabarmati and old Wadaj scenario has been taken from real census data, in design 7m pressure is for low rise area (Bungalow or tenement) as design has been done with census data so detail has been taken from the AMC (Ahmedabad Municipal Council). so, wherever there is 7 m pressure means development is in horizontal way.

In India, for the designing of water supply guideline, we are following the CPHEEO manual on, "Water Supply and Treatment", published by Ministry of Urban Development, New Delhi, constituted by Government of India. To make the system economical and reduce the water loss (With increasing the pressure by 1 unit i.e. 1 to 2 kg/cm2 or 1 to 2 bar, water loss increases by 10%), we have designed the water network in such a manner than residual pressure can be maintained between 7 m to 30m. As per CPHEEO manual in any case the residual pressure should not exceed the 22 m. (30 m is only near to the overhead water tank.)

I agree with you that the residual pressure of 7 m may not be sufficient for supply of water directly to consumers located on second and higher floors, in this case there is need to provide the booster system. Provision of booster system at consumer end is more economical than designing the water network with higher residual pressure.

**2.**Thank you so much for brings our attention on that, we will change and re arrange it such a way so that it can be easily understand and sounds like in a sequence.

"The type of the research is applied research hence, the more weightage has been given to the literature study, literature study has been done on related case studies respectively to the India context. From that Malkapur, Maharashtra- India has been selected for the detail case study.

Then for designing for 24x7 water supply system, data has been collected from the Ahmedabad Municipal Council (AMC) – Water department. Then on basis of given parameter pilot ward has been selected for the 24x7 water design. With help of WaterGEMS software design has been done and from the design proposal has been raised and then on result of the design and selected proposal, economic analysis has been done."

**3**.Your suggestions are well accepted, we will change and shift the paragraph about information and software to the next section. Hours of supply, per capita supply, residual pressure, frequency of supply of existing system, all this will be inserted to this section.

| Criteria | Existing Scenario |
|---|---|
| **Hours of Supply** | 2.5 Hours |
| **Per capita supply** | 140 to 160 LPCD |
| **Frequency of supply** | Daily |
| **Residual pressure** | Varies 7 m to 30 m |
| **Pipe Material** | Ductile Iron & Cast Iron |
| **Mode of Recovery** | Water Tax at 30% of General Tax, Yearly. |

**4.** Thanks for your detail observation. Yes, design has been done with reliable criteria for 24x7 WSS according to CPHEO manual only. WaterGEMS software apart from design for steady state flow, also models the system according to the given pattern of usage of water at the different time of the day in the 24 x 7 availability. This gives the design of the system as a whole i.e. ESR and distribution network. The WaterGEMS software uses Darwin designer which is a generic algorithm. It provides multi criteria optimization. The solutions provided by the software are ranked. This allows the user to choose the best solution which suits to his requirement of pressure and availability of water. Criteria has been used for the design of 24x7 WSS is as follow:

| Criteria | 24x7 WSS |
|---|---|
| **Hours of Supply** | 24 Hours |
| **Per capita supply** | 150 LPCD |
| **Frequency of supply** | Daily |
| **Residual pressure** | Varies 7 m to 30 m |
| **Pipe Material** | HDPE pipes |
| **Mode of Recovery** | Telescopic Tariff |

For strengthening the existing pipe parallelly is not possible throughout the network due to insufficient width especially in street areas as all utility has to go through beneath the road only.

we have taken existing water distribution map from AMC and pasted on ArcGIS and then by using satellite raster image and with help of thiessen polygon method created water boundary for the WaterGEMS software and then started 24x7 design with existing water distribution map and after all these process when run the design it gives all the errors in the distribution network by solving all that we come to the final design of  24x7 WSS for selected wards. From that design we come to know that 85% of the existing network has to be replace for continuous WSS.

Figure 3 & 4 shows the 24x7 WSS design for the Sabarmati and old Wadaj ward which includes the 24x7 hydraulic model of water distribution network with various color codding and shapes where the shapes indicates the water pressure and diameter of the pipes in the distribution network. While the table information reflects the proposed infrastructure from the 24x7 WSS design which justify the no. of Overhead tank required with exact quantity, Diameter and Material of the pipeline.

**5.** Yes you are right the option has been selected for the to 24x7 mode of water supply is costlier among the all 3 given options. And agree that HDPE pipes and trenchless technology has its own disadvantages too. But the scenario in Ahmedabad city is that all the drainage, water, storm water pipes are of Ductile iron pipe and below table is showing how HDPE pipes are far better than existing ductile iron pipes.,

| HDPE Pipes | DI Pipes |
|---|---|
| • High flow characteristic | • Rate of corrosion |
| • Light in weight | • Heavy in weight |
| • Excellent flexibility and strength | • Not that much flexible |
| • Minimize frictional loses | • High friction |
| • Laying length 40' | • Laying length 18',20' |
| • Pressure 260psi | • Pressure 350psi |

All utility pipes and lines pass through the main road network and that too beneath the road line, so if they used trenchless technology, they can save the citizen from suffering lots of traffic jam, noise & air pollution due to excavation.  In advance Ahmedabad city has sandy strata and every year in monsoon city has to face land collapsing especially where the land has been opened for the repair or construction of all these utility lines and pipes.  By keeping all these

things in HDPE and trenchless technology is the best option for the Ahmedabad city by its social, environmental and sustainable point of view.

**6.** Public share contributed by public only. Yes, we will definitely make it in more detail and clear thanks for drag our attention.

Total project cost in 72 crores in which 10% would be paid by the public while remaining would be Government shares. In this case scheme is in ratio of (90:10) in between government and public. Once the project has been laid down to running it successfully government will give it on project basis to the private companies. Which shows that project is PPP mode.

**7.** yes there is some mistake in calculation of table.2 sorry for the inconvenience.

| Sr. No. | Description | Unite No. | Rate Rs. | Total Cost Rs. (Crore) |
|---|---|---|---|---|
| 1 | One-time connection charge | 30,531 households | 2,300.00/House | 7.02 |
| 2 | O & M cost | 30,531 households | 2,400.00/House-Yearly | 7.32 |
| 3 | Average water usage charge | 30,531 households | 3,276.00/House-Yearly | 10.00 |
| **TOTAL REVENUE COST Rs.** | | | | **24.34/ year** |

This final figure shows the revenue would be generated yearly which includes O&M charges, one-time connection charge and telescopic rat of water meter, you have to pay what you have used. This kind of survey also has been carried out in pilot project and its result is very positive.

Design has been done with forecasted population till 2050, so till that there is no issue for increasing in household over the period.

**8.** yes you are right that many apartments type buildings have several households but not with single connection. And if it is so then there is single connection for the electricity meter also, so they have solution for that also same has to be applied for the water meter too.

As I mention in point 5 if you consider all that point and then make economic calculation it would definitely viable for this option. And what we are telling is also a kind of suggestion if you look forward and think as a whole it would be more convenient.

For willingness to pay, telescopic tariff has been calculated on basis of actual consumption of water, where users have to pay on amount of their water usage. which is appreciated by the public also as they have to pay very nominal amount if they had used nominal quantity of

water. It is very helpful to environment also as there is no unnecessarily wastage of water would be happened.

**9.** If anyone really wants to do this, it is possible with help of WaterGEMS software you can easily find which pipe has to be replace and with help of ultra sound system you can find its age and leakage condition and with help of GIS you can find out existing coordinates of that pipes.

Again, thank you so much for your valuable time and detail study of our research work, we are sorry that you don't like it worth, we will try to make it worth by making changes which has been suggested by you for sure in manuscript.

---

## Author Comment (AC2) · 14 Dec 2018

**Responses to the Editor's and Reviewers' Comments**

Thank you so much for your such valuable comments which gives us a Chance to defend and explain our though process in a better way regarding our study.

First of all, this research is purely applied research which can directly implement on real life situation and specified for the Ahmedabad city and its water system. Hence, the literature study has been done purely based on similar kind of case only for India, you can refer cited references at end of the paper, and the case study of Malkapur, Maharatsra-India, which has won the national award for successful implementation of 4x7 water supply system and which has been taken as the bench mark for the study, then only it has been selected; along with this pros and cons of the 24x7 water supply system has been studied from the case studies of Hyderabad, Pimpri-Chinchwad and Nagpur cities of India which has ongoing pilot projects for 24x7 water supply system, which clearly justifying that literature study has been done deeply.

- Ok sure We will rearrange the line with reference. (Line-9)
  As following:

  "Water is also one of the important physical environments of human and has effect on the health and hygiene of mankind."

- Ok, sure we will correct that sentence and remove the word 'Brutal'. (Line-19)
  As following:

  "The present water supply practice is non-confirming to designed hydraulic parameters, and also the system is severely affected insufficient hydraulics, leading to many of the current critical issues which keep the Local Authorities in a difficult situation."

- Thanks for the proposal.  Yes, we can definitely make changes according to your comments and give reference of this sentence to the paper. (Line-35)

- Continuous water supply is one of the important norms for the smart city, we have used the word 'One of the' we never said that continuous water supply is the Only norms. (Line-40)

  "Continuous water supply is also one of the important norms for smart cities. Recently the Government of India has decided to convert 100 cities into smart cities. Ahmedabad has been selected as one of the 100 cities. "

- Yes, we can add 3-4 lines which explain the research methodology (Line-52), which is as below:

  "The type of the research is applied research hence, the more weightage has been given to the literature study, literature study has been done on related case studies respectively to the India context. From that Malkapur, (Maharashtra- India) has been selected for the detail case study. After that for designing of 24x7 water supply system, data has been collected from the Ahmedabad Municipal Council (AMC) – Water department. on basis of given parameter pilot ward has been selected for the 24x7 water design. With help of WaterGEMS software design has been done and from the design proposal has been raised and result of the design and selected proposal, economic analysis has been done."

- Yes, sure we can rearrange Case study (Section III) in a such a way so that it looks like a story telling sequence and not a direct jump. (Line-77-98)

  "The scheme for providing 24x7 water supply system in Malkapur (Maharashtra, India) is unique and remarkable as it has won national award for this scheme for successful implementation and running of 24x7 water supply project. The project was implemented under the guideline of Water supply and sanitation department of Maharashtra - Maharashtra Jeevan Pradhikaran (MJP). By considering its all facts Malkapur has been selected as a detail case study for this applied research."

- In Section IV- study and finding, (Line-102-120) describe the whole scenario of the water supply system in the Ahmedabad city. So, we will add name of the Ahmedabad city so that it easily can understand. (Line-115)

- Yes, last paragraph is the nut shell of the whole study, which is not a result, that's why it is in general statement of the whole research paper to justifying the tittle. Result of the study has been pointed and explained with the 1-10-point number just above this paragraph. (Line-245-247)